# Catalyst-Free Click Chemistry for Engineering Chondroitin Sulfate-Multiarmed PEG Hydrogels for Skin Tissue Engineering

**DOI:** 10.3390/jfb13020045

**Published:** 2022-04-18

**Authors:** Gustavo F. Sousa, Samson Afewerki, Dalton Dittz, Francisco E. P. Santos, Daniele O. Gontijo, Sérgio R. A. Scalzo, Ana L. C. Santos, Lays C. Guimaraes, Ester M. Pereira, Luciola S. Barcelos, Semiramis J. H. Do Monte, Pedro P. G. Guimaraes, Fernanda R. Marciano, Anderson O. Lobo

**Affiliations:** 1LIMAV—Interdisciplinary Laboratory for Advanced Materials, BioMatLab, Materials Science & Engineering Graduate Program, UFPI—Federal University of Piauí, Teresina 64049-550, PI, Brazil; gustavoo.existe@gmail.com; 2Division of Engineering in Medicine, Department of Medicine, Brigham and Women’s Hospital, Harvard Medical School, Boston, MA 02115, USA; 3Division of Health Sciences and Technology, Harvard University—Massachusetts Institute of Technology (MIT), Cambridge, MA 02139, USA; 4Biochemistry and Pharmacology Department, UFPI—Federal University of Piauí, Teresina 64049-550, PI, Brazil; daltondittz@ufpi.edu.br; 5Physics Department, UFPI—Federal University of Piauí, Teresina 64049-550, PI, Brazil; franciscoeroni@gmail.com (F.E.P.S.); marciano@ufpi.edu.br (F.R.M.); 6Department of Physiology and Biophysics, Institute of Biological Sciences, Federal University of Minas Gerais, Belo Horizonte 31270-901, MG, Brazil; daniele.gontijo@yahoo.com.br (D.O.G.); sergiosc1789@gmail.com (S.R.A.S.); anacastro.alc@gmail.com (A.L.C.S.); layscordeiro@gmail.com (L.C.G.); luciolasbarcelos@gmail.com (L.S.B.); ppiresgo@gmail.com (P.P.G.G.); 7Laboratory of Immunogenetics and Molecular Biology, UFPI—Federal University of Piauí, Teresina 64049-550, PI, Brazil; estermpereira@ufpi.edu.br (E.M.P.); semiramis@ufpi.edu.br (S.J.H.D.M.)

**Keywords:** chondroitin sulfate, polyethylene glycol, biomaterial, bioorthogonal chemistry, skin tissue engineering

## Abstract

The quest for an ideal biomaterial perfectly matching the microenvironment of the surrounding tissues and cells is an endless challenge within biomedical research, in addition to integrating this with a facile and sustainable technology for its preparation. Engineering hydrogels through click chemistry would promote the sustainable invention of tailor-made hydrogels. Herein, we disclose a versatile and facile catalyst-free click chemistry for the generation of an innovative hydrogel by combining chondroitin sulfate (CS) and polyethylene glycol (PEG). Various multi-armed PEG-Norbornene (A-PEG-N) with different molecular sizes were investigated to generate crosslinked copolymers with tunable rheological and mechanical properties. The crosslinked and mechanically stable porous hydrogels could be generated by simply mixing the two clickable Tetrazine-CS (TCS) and A-PEG-N components, generating a self-standing hydrogel within minutes. The leading candidate (TCS-8A-PEG-N (40 kD)), based on the mechanical and biocompatibility results, was further employed as a scaffold to improve wound closure and blood flow in vivo. The hydrogel demonstrated not only enhanced blood perfusion and an increased number of blood vessels, but also desirable fibrous matrix orientation and normal collagen deposition. Taken together, these results demonstrate the potential of the hydrogel to improve wound repair and hold promise for in situ skin tissue engineering applications.

## 1. Introduction

The pursuit of ideal biomaterials that perfectly match the building blocks of organs and tissues and that are well recognized and integrated with the microenvironment surrounding the tissues and cells is a never-ending challenge [1,2]. In this context, hydrogels that are three-dimensional (3D) crosslinked polymeric systems with high hydration content and have mechanical and physical properties similar to native extracellular matrix (ECM) represent a promising candidate in this pursuit [3]. Besides great mass transport, they have demonstrated great potential for use in a wide range of biomedical applications, such as cell encapsulation, drug delivery, and scaffolds for tissue engineering [4,5,6]. The use of hydrogels, e.g., injectable material [7,8,9], bioink for 3D bioprinting [10], and nanofiber matrices [11,12], indicate innovative emergent demands in tissue engineering [13,14]. The challenges and limitations to overcome in the development of innovative hydrogels include the lack of universal, versatile, and facile fabrication technologies [2,15]; the use of organic solvents or catalysts that might induce toxicity [16]; few technologies provide on demand and controlled formation of the crosslinked hydrogel that is easy to handle [17,18]; and their tunable mechanical and chemical properties [19,20]. Other challenges in the formation of hydrogels include controlling the gelation time to increase the in situ use [21,22]. As a synthetic approach for hydrogel biomaterials, click chemistry stands out, and the current methodology provides several advantages. Click chemistry is characterized by remarkable selectivity and effectiveness, in addition to allowing the use of different solvents and different functional groups [23,24,25]. Possible chemical reactions for obtaining hydrogels are ionic interactions, hydrogen bonds, condensation reactions, addition reaction, photopolymerization, or enzymatic crosslinking, which are all possible to adapt through click chemistry [26,27]. Moreover, the click chemistry concept combined with techniques such as post-polymerization, also called double crosslinking, offers a viable way to obtain functional surfaces [28].

Various polymers and other materials can be used to produce hydrogels [3]. The process of choosing the precursors allows the control of the desired characteristics in the final product and can be done based on the intended applications [29]. The literature reports different formulations of polyethylene glycol (PEG) [30,31] and chondroitin sulfate (CS) [32,33,34] with the addition of different reactive groups that have shown promising results in the rate of wound contraction [35], histology, and pro-healing in granular tissue [36,37].

Although all the above-mentioned technologies provide excellent tools to use hydrogels for various biomedical applications, there remains room to improve and expand the available scope with new technologies providing for the facile formation of important hydrogels. CS provides the tissue with a good resistance to compression [38] and exhibits anti-inflammatory and anti-catabolic activity [39]. Present in large quantities in native ECM and on the surface of cells, this polymer enhances growth and contributes to neurotrophic factors [40]. CS is an important component in wound healing, promoting cell adhesion and cell proliferation [41,42,43].

Synthetic PEG hydrogels present great characteristics, e.g., in 3D cell culture, where they have few interactions with proteins present in culture media. The cells can grow without interference of the material. PEG gels can be engineered to form a desired structure to mimic some features of the native ECM microenvironment [6].

PEG in the multi-arm form has been used in several applications for tissue engineering and drug delivery and has demonstrated good applicability based on the number of arms. The four- and eight-armed polymers have been shown to promote the gelation time of the hydrogels and result in good elasticity, because the use of multiple arms and high concentrations of precursor solutions provide tunable stiffness and biocompatibility of the hydrogels [44,45].

The synthetic process of many hydrogels involves the use of organic solvents or catalyzed reactions, which could be a problem in the application of the material for tissue engineering, because it increases the cytotoxicity of the final biomaterial [46].

Hydrogels are desirable materials for skin tissue engineering and regeneration, providing an important scaffold that promotes various cellular mechanisms and therefore the increased success of healing. Although a wide range of technologies and products have been developed and employed for the regeneration of skin tissue, there remains room to improve the engineering of sustainable hydrogels generated through a facile and efficient approach. Therefore, we envision that our proposed click chemistry devised hydrogel will be a suitable platform for in situ skin tissue engineering and wound healing applications. Herein, we disclose an innovative, catalyst-free, click-based approach for the engineering of chondroitin sulfate-multiarmed PEG Hydrogels.

## 2. Materials and Methods

### 2.1. Materials

Chondroitin sulfate (CS) sodium salt from shark cartilage, *N*-Hydroxysuccinimide (NHS) (molecular weight (Mw) 115.09 g/mol, purity 98%), 2-(*N*-Morpholino)ethanesulfonic acid (MES) buffer (Mw 195.24 g/mol, purity 99%), *N*-(3-Dimethylaminopropyl)-*N*’-ethylcarbodiimide hydrochloride (EDC) (Mw 191.70 g/mol, purity 98%), Hydroxylamine solution 50 wt% in H_2_O (Mw 33.03 g/mol), and Formalin solution (neutral buffered, 10%) were purchased from Sigma-Aldrich (St. Louis, MO, USA). Tetrazine-Amine (Mw 187.09, purity 95%) was purchased from Click Chemistry Tools (Scottsdale, AZ, USA). We purchased 8-Arm PEG-Norbornene (8A-PEG-N) (Mw 20 kD and 40 kD) and 4A-PEG-N (Mw 20 kD) from Creative PEG Works (Durham, NC, USA). Dialysis membrane (MwCO 3.5 KD) was purchased from Spectrum (Waltham, MA, USA).

### 2.2. Tetrazine Chondroitin Sulfate (TCS) Synthesis

TCS was synthesized according to an established protocol with a small modification [5]. Briefly, 1.0 mol of CS from shark cartilage was dissolved in 50 mmol/L MES buffer (pH 6.5). Subsequently, 52.7 mmol of NHS, 52.1 mmol of EDC, and 5.0 mmol of Tetrazine-amine were mixed and stirred at room temperature for 24 h. Afterwards, the reaction was quenched with hydroxylamine, followed by purification through dialysis for 4 days against deionized water, and then freeze-dried.

### 2.3. Preparation of Click TCS-A-PEG Hydrogel

Click hydrogels were prepared by first separately dissolving freeze-dried TCS and three different armed PEG-Norbornene (8A-PEG-N (Mw 20) and 40 kD; 4A-PEG-N (Mw 20)) to a final concentration of 4 wt% in PBS (1 mol/L, pH 6.9). Then, the precursor solutions were mixed in a 1:1 ratio in ambient conditions to form the gel.

### 2.4. Characterization of Hydrogel Properties

The gelation time was determined through rheological experiments, where the TCS and A-PEG-N polymer solutions were directly pipetted onto the bottom plate of a TA Instruments ARG2 rheometer equipped with a 20-mm flat upper plate geometry. A Peltier base was used to control the constant temperature at 37 °C. Hydrogel samples were subjected to 1% strain at 1 Hz, and the storage moduli (G’) was monitored for 15 min. For Young’s modulus measurements, the click TCS-A-PEG-N hydrogels were formed under home-made siliconized glass plates with 8.0 mm diameter and 1.0 mm height. After 20 min of crosslinking at room temperature, cylindrical discs were punched using an 8 mm biopsy punch, transferred to Dulbecco’s Modified Eagle’s medium (DMEM), and swollen to equilibrium for 24 h at 37 °C. Swollen hydrogel sample dimensions were measured using calipers for volumetric swelling ratio measurements, and then subjected to unconfined compression testing (1.0 mm/min) using a 10 N load cell with no preload (TA.XT plus Texture Analyzer, TA Instruments). Young’s modulus was calculated as the slope of the linear portion (first 10%) of the stress–strain curves. The tests were performed in triplicate.

### 2.5. Scanning Electron Microscopy (SEM) Evaluation

The prepared hydrogels were freeze-dried prior to analysis. The pore sizes and surface morphology of the hydrogels were detected by an FEI Quanta FEG 250 electron scanning microscope (Thermo Fisher, Waltham, MA, USA) at 20.0 kV.

### 2.6. Swelling Behavior Evaluation

The prepared hydrogels were freeze dried prior to analysis. The dried materials were weighed and immersed in PBS (pH 7.4). To obtain the swelling of the hydrogels, the hydrated hydrogels were carefully taken out from the PBS solution and the surface carefully dried by paper towel. The hydrogels were weighted every 10 min for the first hour, and then every 20 min until mass equilibrium occurred. The swelling property (SP) was calculated according to Equation (1):(1)$$SP=\frac{\left(Ws-W0\right)}{W0}$$

In this equation, *W_s_* represents the mass of the swollen hydrogel while the *W*_0_ represents the weight of the dried hydrogel. All the measurements were performed in triplicate and the results are presented by mean ± standard deviation.

### 2.7. Adhesion Assay

Mouse embryonic fibroblast cell line expressing green fluorescent protein (3T3-GFP) was challenged to adhere on the engineered polymers. Briefly, 200 µL of a 1:1 mixture of CS (4 wt%) with PEG (20 kD 4-Arm, 20 kD 8-Arm or 40 kD 8-Arm, 4 wt%) was polymerized in the bottom of a 24-well plate. Ultraviolet light sterilization was performed before the biological assay, for 30 min inside a laminar flow chamber. Subsequently, NIH 3T3-GFP cells (2 × 10^4^ cells/well) were seeded and allowed to adhere for 24 h. Unadhered cells were washed twice with PBS. The tests were performed in triplicate. All images were acquired in an Inverted EPI-fluorescence microscope (EVOSTM M5000, ThermoFischer^®^) at 40× magnitude, and the percentage of adhesion was calculated compared to the control group. Mean ± standard deviation (SD) of each group were compared and considered statistically different when *p* < 0.05 after one-way ANOVA analysis, followed by Tukey’s test (GraphPad Prism 7, San Diego, CA, USA).

### 2.8. Animals

All animal care and experimental procedures complied with the guidelines established by our local Institutional Animal Welfare Committee. The study was approved by the Animal Care Committee guidelines of the Federal University of Minas Gerais (protocol 248/2021). Efforts were made to avoid all unnecessary distress to animals. Eight- to ten-week-old male C57/Bl6 mice were provided by the Center for Animal Care (CEBIO) of the Federal University of Minas Gerais (UFMG), Belo Horizonte, MG, Brazil. After the wounding surgery, the animals were individually housed in ventilated cages (Alesco, Brazil) at 20–24 °C, 50–60% humidity, and 60 air exchanges per hour in the cage. They were fed with standardized mouse chow pellets (Nuvilab CR1, Quimtia S/A, Argentina) and water ad libitum. The light/dark cycle was 12/12 h with lights on at 7:00 am and lights off at 7:00 pm.

### 2.9. Excisional Wound Model and Treatment

Mice were anesthetized intraperitoneally with a mixture of ketamine 100 mg/kg and xylazine 10 mg/kg diluted in saline. Four excisional wounds were performed in the dorsal skin of the animals using a 5-mm diameter biopsy punch and the entire thickness of the skin was removed, as previously described [47]. Immediately after surgery, each wound received 10 mL of the engineered hydrogel or PBS. The wounds were photographed, and their size was determined using a digital caliper immediately after surgery and at days 1, 3, 5, 7, 10, and 14 post-wounding. The results were expressed as closure percentage in relation to the original size (1 − (wound area)/(original wound area) × 100).

### 2.10. Blood Flow Evaluation

Laser Doppler perfusion imaging (LDPI) is a non-invasive technique to assess microcirculation. Microcirculatory changes will affect the wound repair. Blood flow measures of the wound tissue provide color-coded two-dimensional images, in which hot colors represent points with higher blood flow [48]. Blood flow measures were performed in the wound area by LDPI (Moor Instruments, Devon, UK), a non-invasive technique, as previously described [49]. The LDPI was carried out 14 days after wounding surgery in anesthetized mice at the minimal level of ambient light to avoid any influence on the laser light and recorded signals. The animals were kept at a constant temperature of 37 °C for 5 min before and during the imaging procedure. For scanning the injured area, the wound or scar formed after the wound closure was considered as the central point of a 5-mm diameter circumference. The mean pixel value of each scanned image was calculated using the MoorLDI V5.3 software (Moor Instruments, Axminster, UK), and the calculated mean flux was expressed as relative units, which represent the average blood flow of the injured skin.

### 2.11. Histological Analysis

Wound tissues were harvested from mice and fixed in a 10% neutral buffered formalin solution for 48 h, embedded in paraffin, and finally cut into sections 10 μm thick. Histological sections were stained with hematoxylin and eosin (H&E). Morphological analysis of the blood vessels and orientation of fibrous matrix was performed using a microscopy (Olympus, CX41) at magnification of 400×. After morphometric analysis, the number of vessels were counted on 10 randomly chosen fields per slide, as previously described [50].

### 2.12. Collagen Deposition Evaluation

Collagen production was assessed using Picrosirius red stain [51]. This method allows differentiation between thick/mature collagen (red and orange to yellow birefringence) and thin/immature collagen (greenish birefringence), under polarized light, according to the degree of matrix deposition and maturity [51]. Briefly, samples were fixed in a 10% neutral buffered formalin solution for 48 h and then processed for paraffin inclusion. After cross-section to 10 μm thickness, a deparaffinization step was performed in xylene and ethanol followed by hydration in a series of graded alcohols until distilled water, followed by incubation with a Sirius red solution diluted in 0.1% saturated picric acid. After 45 min at room temperature, samples were rinsed with distilled water. Sections were examined by polarization microscopy (Olympus, CX41). To quantify each collagen area type, a threshold algorithm was used to determine the red (red > green + blue * 1.2) and green (green > red + blue * 0.7) pixels (the multiplication factor in the formulae was determined empirically for the current image set). Then, the red and green pixels were counted to determine the tight and thin collagen areas, respectively. To obtain the tissue area, the brightfield image was transformed into a grayscale image and binarized [52]. In an attempt to segment the tissue of interest and avoid artifacts and segmentations outside of the region of interest, a combination of the Canny edge detector and morphological operations (dilation and erosion) were used [53]. The total tissue area was quantified by counting the number of white pixels inside the region of interest of the binarized image. Once the pixel area of each collagen type and tissue was obtained, the collagen content was calculated as a percentage of the area tissue.

### 2.13. Statistical Analyses

Analyses were performed using the GraphPad Prism 8.0 software (GraphPad Software, Inc., San Diego, CA, USA). Results were presented as mean ± SEM. Two-way ANOVA was used for graph lines to verify the interaction between the independent variables time and treatment to analyze the wound closure rate, followed by a Bonferroni posttest. Comparisons between the two groups for blood flow analysis and number of blood vessels were carried out using the Student *t*-test for unpaired data. A *p* value < 0.05 was considered significant.

## 3. Results and Discussion

To prepare clickable Tetrazine-CS (TCS) polymer, the tetrazine moiety was introduced to high Mw CS biopolymer using conventional carbodiimide chemistry through the addition of EDC and NHS in MES buffer at room temperature for 24 h, as illustrated in Figure 1a [5,6]. The reaction was finished by quenching with hydroxylamine, followed by eliminating the unreacted byproducts through dialysis for four days with 12–14 kD membrane. The success of synthesizing the TCS polymer was disclosed in our recent publication through various characterization methods [54].

After the synthesis of the clickable TCS, the multi-arm PEG-Norbornene (A-PEG-N) was simply mixed in 1:1 ratio to form a spontaneous and stable hydrogel via an inverse electron demand Diels–Alder reaction (IEDDA), in the absence of any catalyst (Figure 1b). The chemical crosslinking promotes the formation of covalent bonds between the tetrazine and norbornene moieties, initiating the click chemistry, validated by the spontaneous formation of a self-standing hydrogel within minutes (Figure 1c) [5,41]. Various A-PEG-N with varying Mw and number of arms (4A-PEG-N (Mw = 20 kD) and 8A-PEG-N (Mw = 20 kD and 40 kD)) crosslinked with TCS were evaluated and all the groups tested provided the spontaneous formation of hydrogels that displayed good stability, resistance, and the same shade of pink (Figure 1c).

The crosslinking time for the hydrogel formation of the polymers was investigated through rheological characterization by time sweep experiments. Rheological evaluation was employed to precisely determine the gelation time point, where it was extracted from the curve of storage modulus (G′) versus time graph (Figure 2a,b). Time sweep experiments for all the samples were conducted at 37 °C with 1% strain at 1 Hz and the gelation time was indicated at the time point where the storage modulus (G′) started to increase. TCS-4A-PEG-N (20 kD) had a gelation time at 6.6 min, TCS-8A-PEG-N (40 kD) at 7.2 min, and the TCS-8A-PEG-N (20 kD) at 3.9 min. The differences between the gelation time of the various hydrogels stems from the differences in Mw and number of arms on the PEG-N groups. The results suggest that lower Mw and more arms on the PEG-N promotes faster gelation time (Figure 2a,b).

To determine the mechanical behavior of the hydrogels, an unconfined uniaxial compression test was performed, and the Young’s modulus was recorded to determine the elasticity of the crosslinked hydrogels. The mechanical properties of the ECM have been shown to affect fate and function of cells in 2D and 3D environments [55,56]. Prior to measurements, the various hydrogels were crosslinked in a customized mold to generate disc shaped hydrogels. These discs were swollen for 24 h in a culture medium. Afterwards, the crosslinked hydrogel samples were subjected to unconfined compression tests providing the strain–stress curves (Figure 2c). The compressive Young’s modulus was calculated (Figure 2d). The values of the Young’s modulus were 145.93 Pa (TCS-4A-PEG-N (20 kD)), 145.96 Pa (TCS-8A-PEG-N (20 kD)), and 112.18 Pa (TCS-8A-PEG-N (40 kD)) for the respective hydrogel formulations. Although there were no statistically differences between the various formulations, it is possible to perceive that the largest chain size resulted in the decrease of the Young’s modulus, and thus increased elasticity of the hydrogel. Furthermore, varying the number of arms did not produce any significant difference in the mechanical behavior. Instead, the size of polymers had the greatest impact (Figure 2c,d).

The porous structure of materials exerts a major influence on cells and their behavior for wound repair applications, where the pore size and their interconnected structures play an important role [57]. These features allow water uptake and promote cell adhesion on the structural scaffold. Figure 3a,b demonstrates the morphology of the TCS-8A-PEG-N 40 kD formulation displaying a fibrous morphology with porous network. The various formulations displayed different porous sized structures, where the TCS-8A-PEG-N 20 kD hydrogel demonstrated highest porosity with 110 μm, followed by TCS-4A-PEG-N 20 kD (72 μm) and the TCS-8A-PEG-N 40 kD (66 μm) (Figure 3c). Furthermore, the swelling properties of hydrogels are important features in wound repair applications, allowing a large amount of water to be absorbed by the material [58]. The swelling behavior of the devised hydrogels was investigated following the gravimetric analysis method. The TCS-4A-PEG-N 20 kD hydrogel formulation displayed the highest swelling behavior, followed by TCS-8A-PEG-N 20 kD, and finally the TCS-8A-PEG-N 40 kD hydrogel (Figure 3d).

To assess the ability of the engineered hydrogels to adhere to fibroblast cells and viability, the various hydrogel formulations were exposed to NIH 3T3-GFP cells. Cells (2 × 10^4^ cells/well) were grown on the hydrogel surface for 24 h in a controlled atmosphere. Only the live cells were considered adhered. The dead cells were discarded and washed away. Then, the total number of live cells was determined, as demonstrated in the generated photomicrographs (Figure 4a), and the viability was determined (Figure 4b). The increase in the size and a greater number of arms in the A-PEG-N promoted enhanced adhesion of cells onto the hydrogels. Previous studies demonstrated that porosity is an important factor to promote cells adhesion, proliferation, and migration, and the more space available in the hydrogel network also promotes the necessary support to cells [59,60,61,62]. The observed results from the mechanical properties suggest that a lower value of Young’s modulus positively influences the adhesion of the cells. The TCS-8A-PEG-N with the Mw of 40 kD is the most elastic hydrogel and displayed the best results for the adhesion. The TCS-4A-PEG-N and TCS-8A-PEG-N with the Mw of 20 kD exhibited no statical differences in elasticity and similar behavior for the adhesion ability. It is important to note that the TCS-8A-PEG-N with the Mw of 40 kD demonstrated 100% of adhered and viable cells after 24 h (Figure 4b). The compatibility of the devised hydrogels corroborates our previous report on the compatibility of the hydrogels through gene expression evaluations [54].

After demonstrating the feasibility of the TCS-8A-PEG-N (40 kD) hydrogel formulation, we investigated whether this material could improve wound closure and blood flow in vivo, and therefore represent a potential hydrogel candidate for skin tissue engineering and regeneration. Although several reports have disclosed the use of click chemistry strategies for the generation of hydrogels for skin tissue engineering applications, several of these approaches have limitations, such as the use of an activator (e.g., UV) and the presence of a crosslinker, which may cause toxicity to cells [63]. Therefore, ideally, a hydrogel would preferably be free from any activator or catalyst with the option to be generated on demand.

In this report, wounds from both control and treated group (TCS-8A-PEG-N (40 kD)) closed at a similar rate in about two weeks (Figure 5a,b). However, note that wounds can heal spontaneously in this animal model, and future studies will investigate the wound repair using this engineered hydrogel, in diabetic mice. Importantly, the engineered hydrogel improved blood perfusion, measured via scanning laser Doppler, in the region of the injury as observed at day 14, when animals were euthanized (Figure 5c,d). The enhanced blood perfusion is indicative of angiogenesis and an improvement of wound repair [48]. Therefore, increased blood perfusion is critical, e.g., to induce wound repair in diabetic mice [64]. In addition, morphometric analysis found an increase in the number and diameter of blood vessels for the TCS-8A-PEG-N-treated group compared to the control group (Figure 5e,f). The greater number of blood vessels is an indication of angiogenesis, which is one hallmark of the wound repair [64]. Collectively, these results suggest that the TCS-8A-PEG-N (40 kD) hydrogel improves tissue perfusion and angiogenesis, which are critical parameters for wound repair.

Besides changes in blood perfusion and angiogenesis, we assessed the orientation of fibrous matrix, which is one of the most significant differences between normal repair and scar formation [65]. A basketweave-like network orientation was observed for the TCS-8A-PEG-N (40 kD) treated group, while organized fibers in a parallel way were observed for the PBS samples (Figure 6a,b) [65]. Organized and parallel fibers are associated with abnormal scar formation while a basketweave-like network is related to normal tissue [65]. Therefore, our results suggest that TCS-8A-PEG-N (40 kD) induced a normal wound repair compared to PBS.

Next, we assessed the effect of the TCS-8A-PEG-N (40 kD) hydrogel on the collagen deposition, using picrosirius red staining. The excessive collagen deposition and tight:thin collagen ratio above 6:1 can promote abnormal wound repair and different types of scar formations [65]. Our data did not show any differences in total, tight, and thin collagen production between the treated and control groups (Figure 6c,d). Furthermore, the tight:thin collagen ratio was around 5:1 for both groups, which suggested normal repair for treated and control groups (Figure 6d). Further studies in diabetic mice will investigate the scar formation after treatment with this engineered hydrogel.

## 4. Conclusions

The facile and versatile click chemistry-based preparation of the fibrous hydrogels in the absence of any catalyst provided a self-standing hydrogel within minutes by simply mixing the two components, TCS and A-PEG-N. The sustainably prepared hydrogels displayed a tunable gelation time where this parameter could be controlled by varying the Mw and number of arms on the PEG-N groups, where the initial experiments demonstrated that a lower Mw and greater number of arms supports the gelation time. Furthermore, the mechanical properties of the engineered porous hydrogels could be improved by altering the size of the polymers, where a higher Mw provided increased elasticity.

The biocompatibility experiments performed using the cell adhesion assay demonstrated that the most elastic hydrogel formulation, TCS-8A-PEG-N (40 kD), displayed the greatest cell compatibility with no sign of toxicity. This optimal hydrogel formulation further demonstrated not only enhanced blood perfusion and an increased number of blood vessels, but also a desirable fibrous matrix orientation and normal collagen deposition in an in vivo wound closure experiment. Taken together, these results demonstrated the potential of the engineered hydrogel for wound repair.

In conclusion, the initial evaluation of the engineered hydrogels using eco-friendly click technology and the in vivo results demonstrate the potential of the hydrogels as candidates for in situ application in skin tissue engineering and regenerative challenges. Nevertheless, further thorough evaluation with a wider polymer candidate needs to be conducted to provide a solid and clear understanding of the in vivo behavior of the material.

## Figures and Tables

**Figure 1 jfb-13-00045-f001:**
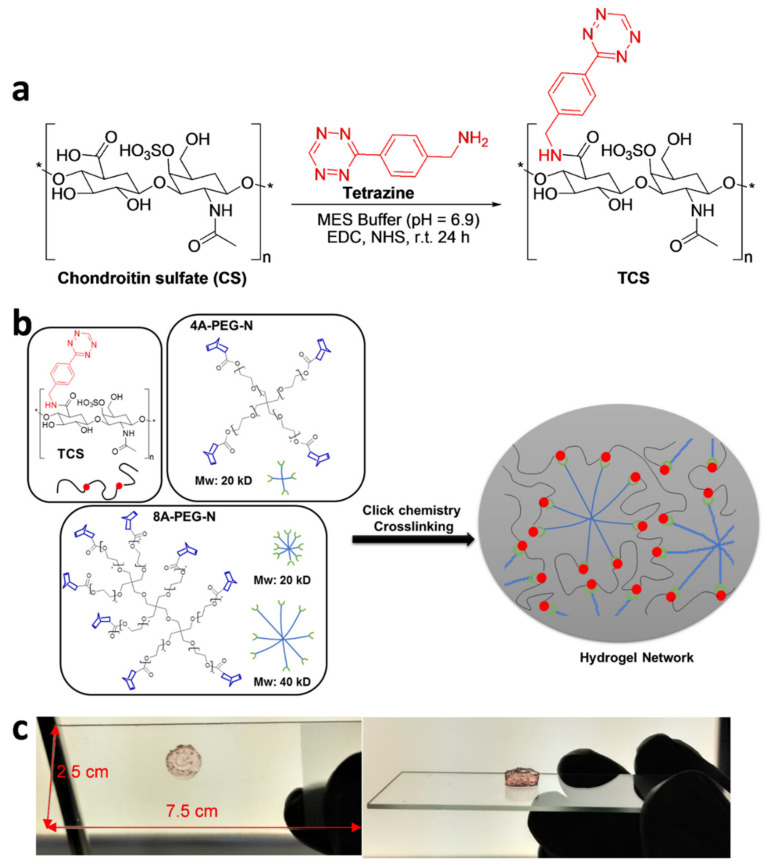
Gelation mechanism based on click chemistry strategy. (**a**) The TCS was prepared through EDC, NHS coupling strategy in MES buffer solution, at 25 °C for 24 h. * represents a fragment of the molecules. (**b**) Schematic representation of spontaneous hydrogel formation mechanism through chemical crosslinking reaction between tetrazine and norbornene moieties within the TCS and A4 (and 8A)-PEG-N groups through click chemistry strategy. (**c**) Photos of the crosslinked and self-standing hydrogel formed within a few minutes of mixing. TCS = Tetrazine chondroitin sulfate; EDC = *N*-(3-Dimethylaminopropyl)-*N*′-ethylcarbodiimide hydrochloride; NHS = *N*-Hydroxysuccinimide; MES = *N*-Hydroxysuccinimide; A4-PEG-N = 4-armed polyethylene glycol norbornene; 8A-PEG-N = 8-armed polyethylene glycol norbornene.

**Figure 2 jfb-13-00045-f002:**
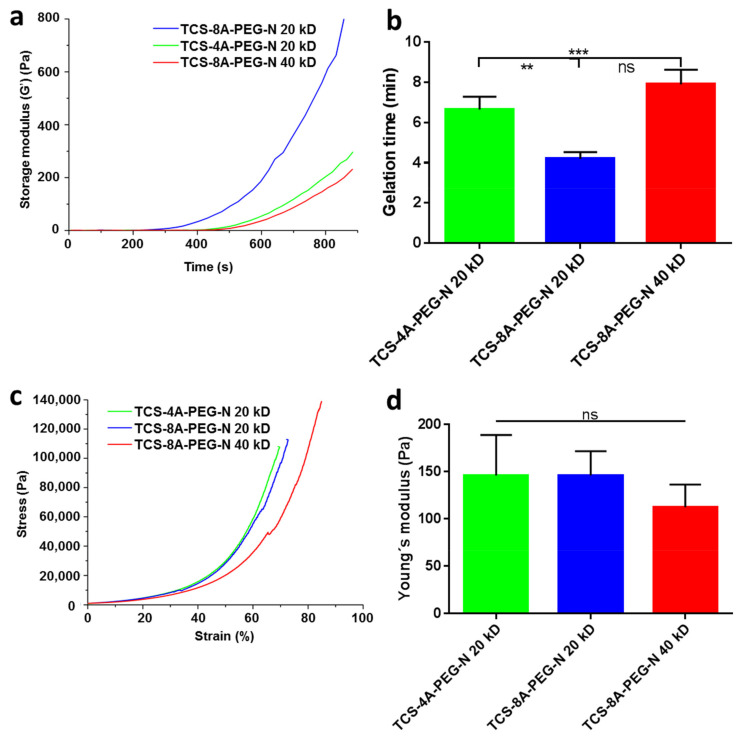
Characterization of the gelation time through rheological experiments and mechanical behavior of the various hydrogel formulations [TCS-4A-PEG-N (20 kD), TCS-8A-PEG-N (20 kD), and TCS-8A-PEG-N (40 kD)] through compression tests. (**a**) Time sweep measurement for the gelation process of 4 wt% solutions by a rheometer (1% strain, 1 Hz, 37 °C). (**b**) Bar graph presenting the gelation time of the various hydrogel formulation. (**c**) Strain-stress curves of the different hydrogels. (**d**) Values of Young’s modulus of all the samples tested. Data are expressed as mean ± SD, *N* = 3, one-way ANOVA, Tukey’s posttest, (**) *p* < 0.01, (***) *p* < 0.005, ns = not significant mean statistical differences. TCS-4A-PEG-N = Tetrazine chondroitin sulfate 4-armed polyethylene glycol norbornene; TCS-8A-PEG-N = Tetrazine chondroitin sulfate 8-armed polyethylene glycol norbornene; kDa = Kilo dalton.

**Figure 3 jfb-13-00045-f003:**
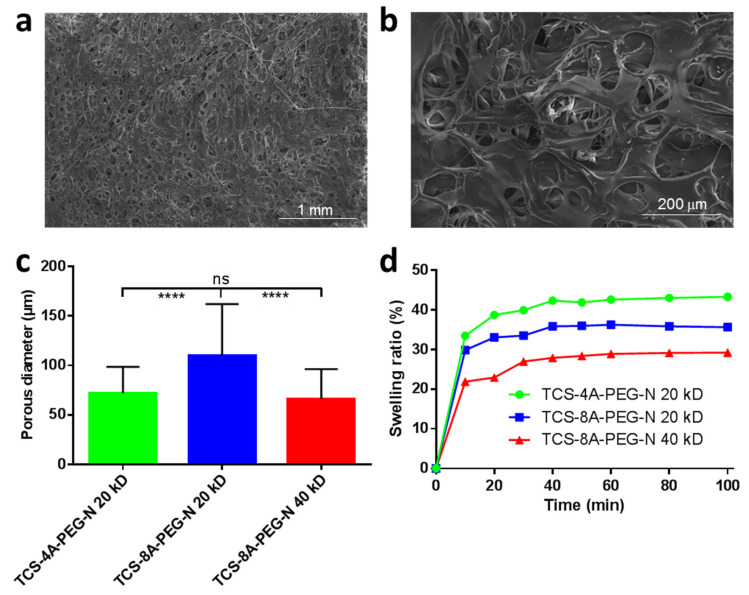
Morphological evaluation and swelling study of the various hydrogel formulations. (**a**), (**b**) SEM images of the TCS-8A-PEG-N 40 kD formulation at scale bar 1 mm and 200 μm. (**c**) Bar graph presenting the porous diameter of the various hydrogel formulations. (**d**) The swelling behavior of the different hydrogels. Data are expressed as mean ± SD, *N* = 3, one-way ANOVA, Tukey’s posttest, (****) *p* < 0.001, ns = not significant mean statistical differences. SEM = Scanning electron microscopy; TCS-4A-PEG-N = Tetrazine chondroitin sulfate 4-armed polyethylene glycol norbornene; TCS-8A-PEG-N = Tetrazine chondroitin sulfate 8-armed polyethylene glycol norbornene; kDa = Kilo dalton.

**Figure 4 jfb-13-00045-f004:**
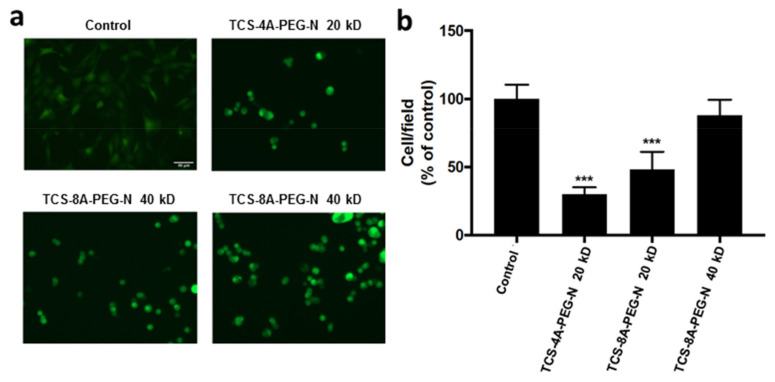
Cell adhesion and viability evaluations on NIH 3T3-GFP fibroblasts seeded onto 4 wt% of the various hydrogel formulations [TCS-4A-PEG-N (20 kD), TCS-8A-PEG-N (20 kD) and TCS-8A-PEG-N (40 kD)] and allowed to adhere for 24 h. (**a**) Fluorescence images of cells adhered to various hydrogels. Individual fluorescent cells were counted and expressed as a percentage of control. Scale bar = 40 µm. (**b**) Cell viability data expressed as mean ± SD, *N* = 3. *** *p* < 0.005, one-way ANOVA, Tukey’s posttest. NIH 3T3-GFP = Mouse embryonic fibroblast cell line expressing green fluorescent protein; TCS-4A-PEG-N = Tetrazine chondroitin sulfate 4-armed polyethylene glycol norbornene; TCS-8A-PEG-N = Tetrazine chondroitin sulfate 8-armed polyethylene glycol norbornene; kDa = Kilo dalton.

**Figure 5 jfb-13-00045-f005:**
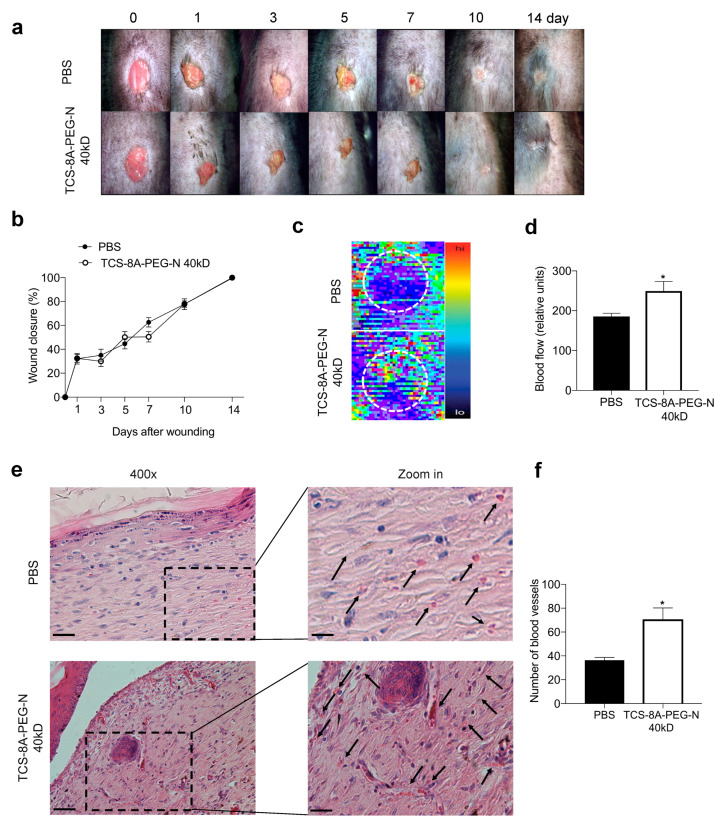
Wound closure rate and blood flow. (**a**) Representative pictures of the wounds. (**b**) Time-course of wound closure during the 14-days experimental period. (**c**) Representative blood flow images. Hot colors represent points of higher blood flow. (**d**) Wound blood perfusion was assessed at 14 days post-wounding. (**e**) Representative images of H&E staining of the wound showing blood vessels (black arrows). (**f**) Number of blood vessels in the wounds after treatment with PBS or TCS-8A-PEG-N 40kD. Data are expressed as mean ± SEM; *N* = 5–7 mice/group. Magnification: 400×, scale bar = 5 mm * *p* < 0.05 vs. control (PBS) group.

**Figure 6 jfb-13-00045-f006:**
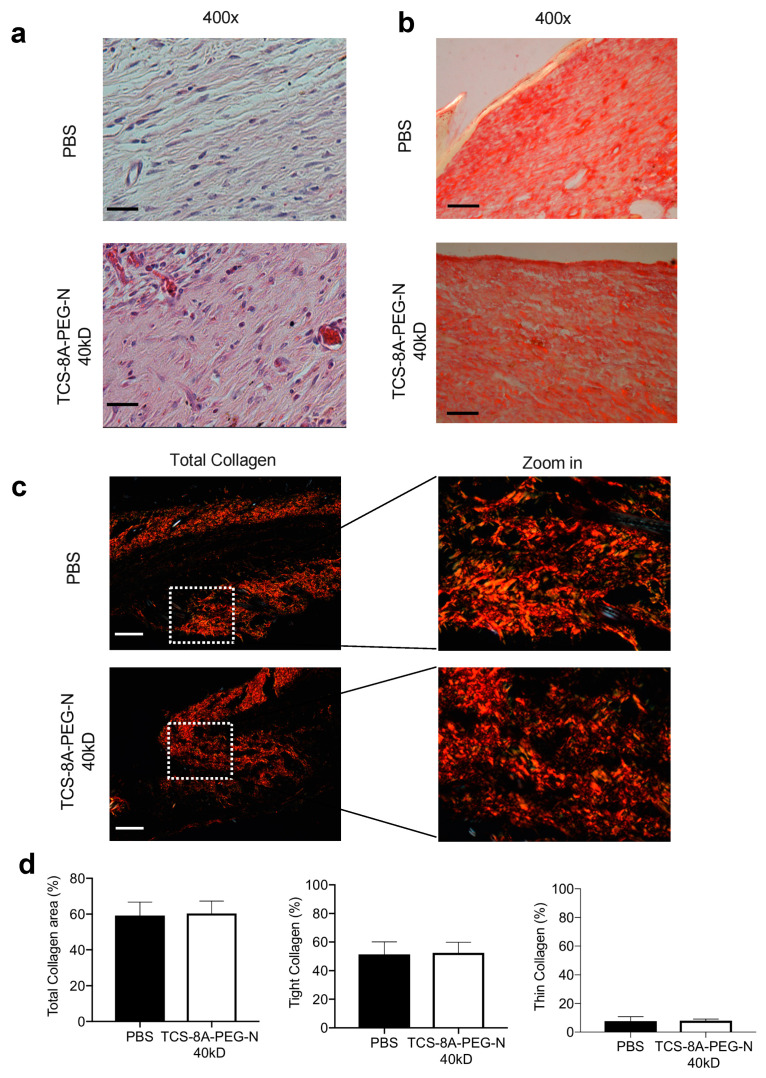
Effects of the engineered hydrogel on tissue repair and collagen deposition. (**a**) Representative images of (**a**) H&E, and (**b**) picrosirius staining showing the orientation of the fibrous matrix at 14 days after treatment (Magnification: 400×, scale bar = 5 mm). (**c**) Representative picrosirius red staining for collagen fiber types viewed in polarized light stained at day 14 after treatment. Objective = 10×; Scale bar = 2 mm. (**d**) Quantification by image analysis of collagen fiber types. Data are expressed as mean ± SEM; *N* = 5 mice/group.

## Data Availability

The data presented in this study are available in the manuscript itself.

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
