# Peer review of "Catalyst-Free Click Chemistry for Engineering Chondroitin Sulfate-Multiarmed PEG Hydrogels for Skin Tissue Engineering"

_jfb, 2022, doi:10.3390/jfb13020045_

Round 1
Reviewer 1 Report
The article “Catalyst-Free Click Chemistry for Engineering Chondroitin sulfate-Multiarmed PEG Hydrogels for Skin Tissue Engineering”, Gustavo F. Sousa, Dalton Dittz, Samson Afewerki, Daniela de Oliveira Gontijo, Francisco E. P. Santos, Luciola Silva Barcelos, Sérgio Ricardo Aluotto Scalzo, Ana Luiza Castro, Lays Cordeiro Guimaraes, Pedro P. G. 5 Guimaraes, Fernanda R. Marciano and Anderson O. Lobo is dedicated to engineered hydrogels by combining chondroitin sulfate and polyethylene glycol.
The article is well written and comprises a lot of analyses.
However, if possible, I would suggest to the authors to insert some microscopical images of obtained hydrogels. From my opinion, this will add value to the manuscript.
Reviewer 2 Report
Sousa et al. reported a novel chondroitin sulfate-multiarmed PEG hydrogel using catalyst-free click chemistry for skin tissue engineering. The results are interesting. I recommend it for publication in JFB after the following issues are addressed.
- Line 97-111, these sentences should be removed.
- What is the molecular weight of CS used in this study?
- The resolution of figure 1, 2, 4, and 5 should be improved to a publishable level.
- Why the TCS-8A-PEG-N (40 kD) hydrogel showed better biocompatibility?
- Line 40-45, Several recent studies (doi.org/10.1016/j.actbio.2021.11.036; 10.1126/science.aay8276; 10.1021/acs.biomac.1c00719) should be included to support such claim.
Reviewer 3 Report
The manuscript refers to Catalyst-Free Click Chemistry to produce hydrogels that are promising for wound healing applications. The composition and fabrication technology used is stated as a controllable and eco-friendly methodology to produce hydrogels. Despite the good mechanical characterization of the hydrogel, the biological characterization needs to be improved. There is a huge gap between the analysis of the mechanical properties and the final-end application in an in vivo model considering the poor characterization of cell viability and toxicity. Moreover, there is a lack of analysis of all the results that can be extracted from the animal model, as more detailed characterization of the scar and the wound bed could be provided. In general terms, I recommend the authors to consider reformulating the final goal of this paper. In this version, this paper is devoted to proving how the hydrogel generated using this technology can promote wound healing, whereas the final conclusion of the paper is that it did not. The hydrogels described in this paper seem to have very good properties that can be characterized more in detail (cell viability and hydrogel degradation) to end up with the prove that when implanted in the wound bed they do not produce an aberrant collagen deposition while they promote blood perfusion. I recommend major revisions.
Abstract
It should specify more clearly the results, giving quantitative data from the results.
Introduction
Line 45: very long sentence difficult to understand. In this part are listed many limitations and challenges of the hydrogels with few references.
Line 60: not well written, consider reformulating the whole sentence.
Line 66: Is stated that in the literature there are several formulations, but there is only one reference specified. Is necessary to specify more clearly which are the formulations found in the literature, which are their limitations and in which stent your technology improves it.
Line 76: what is naturally? Autologous?
Line 76-79: very long, difficult to understand.
Materials and Methods
The number of repetitions for each experiment/condition is not clearly specified.
Line 150: adhesion experiment can only give you information about the substrate with respect to the cell, but this cannot be directly linked to cell viability/toxicity without analyzing cell viability/toxicity using test specifically design for that purpose.
Line 180: Is not well described how this technology works and how it helps in analyzing blood perfusion. Additionally, as it is a non-invasive technology, instead of using it at the end of the experiment, a time-laps analysis at different time points may help to understand how blood invasion occurs. To analyze the endpoint of the experiment when the wound is closed consider including an immune/histological analysis of the scar tissue showing blood vessel arrangement for all the conditions.
Results
In general terms, the quality of the images is very poor. Is not possible to distinguish important details that help to understand the results.
268: Again, cell adhesion cannot be directly linked to the cell viability and toxicity. Extra studies on cell viability (live/dead) and cell toxicity (CellTox Green Cytotoxicity Assay or similar) should be included to complete the viability characterization. Moreover, when a wound closes the fibroblast present in the surrounding healthy tissue invade the wound bed. For this reason, is important to analyze not only substrata adherence but cell encapsulation within the matrix. With that purpose, consider the analysis of cell viability within the hydrogel as for example with Live Dead or AlamarBlue. Although is very well justified why cells adhere to the surface, considering the stiffness and porosity of the hydrogel, this is not a prove of the viability/toxicity.
302: This result is straightforward considering the reported results. Figure 4D has very poor quality and the heat map is not clear. There is a lack of information regarding this technology and how to understand and interpret the results. Moreover, the quality of the images is very low, and much information is lost.
Results regarding collagen deposition should be reformulated. First, the quality of the images makes no possible to distinguish between the different channels and conditions. Despite the quantification shown in the graph, higher magnifications of the images with higher resolution may help to clearly see the collagen disposition. Then the presentation of the results should be considered, as is not clearly stated neither in the M&M nor in the results that you want to analyze that both conditions have the same pattern. With these results seems that you are looking for a difference between conditions but finally you ended in a no-difference condition that was, instead a good point (as you later state). In addition to that, an H&E staining could help to understand the structure of the scar at both culture conditions to see if there is any difference in the tissue (and not only the collagen deposition).
Another important question to answer is, where is the hydrogel when the wound is closed? Is it degraded? Is it integrated within the tissue?
Conclusion
While the conclusion extracted from the mechanical point of view (characterization of the material) are good and they are supported by the results shown, those from the biological part are not.
Line 345: is stated that there is not sign of toxicity, and that is not proven in the results. The result clearly shown that adhesion decreases in the presence of the material. Nonetheless, this adhesivity is not a prove of the non-toxic effect of the matrix.
Line 348: is stated that this hydrogel promotes wound healing, and this is not proven by the results. In the results section the authors show that there is no difference regarding wound healing time nor collagen deposition when the hydrogel is used, and here is stated that it promotes wound healing.
The results from image 5 could be supplemented with the H&E analysis, as it would help to analyze the structure.
Figures:
In general, very poor quality of the images.
Figure1: no scale bar.
Figure 2 (b/d): consider including the statistical significance.
Figure 3: (a) the focus is not clear and the show a very light fluorescence. In b,c and d) they are out of the focus.
Figure 4 (a): there is no scale bars.
Round 2
Reviewer 3 Report
The suggestions made in the previous report has been considered and rectified or sucessfully answered.